# The Balkan Region and the “Nano Gap”: An Underexplored Dimension of In Vitro Biotechnology for Woody Plants

**DOI:** 10.3390/plants14223499

**Published:** 2025-11-16

**Authors:** Valbona Sota, Slađana Jevremović, Eleni Abraham, Vanja Daničić, Dejan Bošnjak, Lilyana Nacheva, Branislav Cvjetković, Vlatko Andonovski, Sanja Bogunović, Efigjeni Kongjika, Svjetlana Zeljković, Darko Jevremović, Zvjezdana Marković, Vladislava Galović, Tatjana Vujović

**Affiliations:** 1Department of Biotechnology, Faculty of Natural Sciences, University of Tirana, 1000 Tirana, Albania; 2Research Center of Biotechnology and Genetics, Academy of Sciences of Albania, 1000 Tirana, Albania; kongjikaef@yahoo.com; 3Institute for Biological Research “Siniša Stanković”—National Institute of Republic of Serbia, University of Belgrade, 11000 Belgrade, Serbia; sladja@ibiss.bg.ac.rs; 4School of Agriculture, Forestry and Natural Environment, Aristotle University of Thessaloniki, 541 24 Thessaloniki, Greece; eabraham@for.auth.gr; 5Faculty of Forestry, University of Banja Luka, 78000 Banja Luka, Bosnia and Herzegovina; vanja.danicic@sf.unibl.org (V.D.); branislav.cvjetkovic@sf.unibl.org (B.C.); 6Faculty of Agrobiotechnical Sciences Osijek, Josip Juraj Strossmayer University of Osijek, HR-31000 Osijek, Croatia; dbosnjak@fazos.hr; 7Fruit Growing Institute, Agricultural Academy, 4004 Plovdiv, Bulgaria; lilyn@abv.bg; 8Hans Em Faculty of Forest Sciences, Landscape Architecture and Environmental Engineering, University Ss. Cyril and Methodius in Skopje, 1000 Skopje, North Macedonia; vandonovski5@gmail.com; 9Department of Genetics, Forest Tree Breeding and Seed Science, Croatian Forest Research Institute, 10450 Jastrebarsko, Croatia; sanjam@sumins.hr; 10Faculty of Agriculture, University of Banja Luka, 78000 Banja Luka, Bosnia and Herzegovina; svjetlana.zeljkovic@agro.unibl.org; 11Fruit Research Institute, 32000 Čačak, Serbia; darkoj@ftn.kg.ac.rs; 12Department of Viticulture and Enology, Faculty of Agriculture, University of Zagreb, 10000 Zagreb, Croatia; zmarkovic@agr.hr; 13Institute of Lowland Forestry and Environment, University of Novi Sad, 21000 Novi Sad, Serbia; vladislava.galovic@gmail.com

**Keywords:** Balkan countries, nanotechnology, innovative tools, micropropagation, in vitro conservation

## Abstract

Although nanotechnology is increasingly applied in plant tissue culture in many parts of Europe, its use in the Balkans remains limited, revealing a regional gap with untapped potential for advancing in vitro propagation and preservation of woody plant species. Building upon a recently published regional review covering 2001–2024, which analyzed in vitro biotechnology progress in nine Balkan countries, this paper introduces the concept of a “nano gap”, referring to the limited connection between existing nanotechnology research potential and its use in in vitro woody plant biotechnology. In Serbia, Greece, Bulgaria, Croatia, and Albania, significant progress has been made in optimizing micropropagation and in vitro conservation strategies by introducing temporary immersion systems, synthetic seed technology, adapting genotype-specific sterilization and multiplication protocols, and modifying established cryopreservation methods for regional woody species. However, the integration of nanotechnology into these systems remains largely unexplored. To date, there are no published results or validated applications for nano-enhanced media or nanoscale delivery systems for micropropagation and in vitro conservation of woody species. The limited integration of nanotechnology may be due to insufficient funding, lack of specialized infrastructure, and limited interdisciplinary expertise. Nevertheless, many Balkan countries possess growing capacities in nano-applications within agriculture and environmental sciences and are ready to advance toward interdisciplinary research and innovation. By mapping both scientific readiness and structural barriers, this review provides a strategic framework for bridging the “nano gap” and offers a novel regional perspective with broader implications for European research policy, sustainable agriculture, biodiversity preservation, and green innovation.

## 1. Introduction

In the Balkan region, in vitro plant biotechnology has developed considerably over the last twenty-five years for woody species of horticultural and forestry interest. A recent regional assessment [1], which analyzed the progress of in vitro plant biotechnology advancements in nine Balkan countries (Albania, Bosnia and Herzegovina, Bulgaria, Croatia, Greece, Kosovo, Montenegro, North Macedonia, Serbia) at the beginning of the 21st century (2001–2024), underscores notable advancements in traditional tissue culture methods for woody species. Countries like Serbia, Greece, Bulgaria, Croatia, and Albania have made significant progress in improving micropropagation protocols and in vitro conservation strategies for many horticultural crops (such as cultivated and spontaneous species of *Prunus* sp., *Pyrus* sp., *Malus* sp., *Citrus* sp., berry fruits, nuts, and grapes) and forest species (such as *Aesculus* sp., *Populus* sp., *Pinus* sp., and others). Several issues have been addressed in these efforts, including species recalcitrance, high contamination rates, and low regeneration efficiency, which are considered persistent obstacles in woody plant tissue culture [2,3].

Despite these advances, the integration of nanotechnology into in vitro systems for woody species has not yet been observed in the Balkan region. To date, there are no published results on nano-enhanced media or nanoscale delivery systems in this area for the Balkan countries, which are the focus of this review [1]. Given the ecological and economic importance of these species, the efficient propagation and conservation of these species remains critical for the development of sustainable agricultural systems and the achievement of biodiversity goals. Globally, nanotechnology is well-integrated into agriculture, improving crop protection and productivity [4,5,6]. Given the significant agricultural and commercial role of in vitro techniques as a complementary plant production system [7,8], scientific research on the integration of nanotechnology and biotechnological plant production has progressed in recent years [9,10,11,12]. The unique physicochemical properties of nanoparticles (NPs), including their large surface-to-volume ratio and controlled release capabilities, make them ideal for various in vitro biotechnological applications in plants [13,14,15]. Some of the commonly used nanoparticles (NPs) in agricultural and biotechnological plant production are AgNPs, ZnONPs, CuONPs, Fe_2_O_3_ NPs, Al_2_O_3_ NPs, Mn_2_O_3_NPs, and many more [16,17,18]. In addition to nanoparticles, nanomaterials such as carbon nanotubes [19,20,21,22] or titanium sulfide nanoribbons [23] are also frequently used in in vitro plant biotechnology.

Such applications are well-documented in herbaceous plant species [24,25,26,27,28], while far fewer studies are available for woody plants. The main reasons for the limited development of nano-enabled protocols for woody plants may be related to the longer life cycles of these species and their recalcitrance [2,29]. Nevertheless, worldwide research in the field of nanotechnology for woody plant in vitro systems has expanded significantly in recent decades [30].

The situation in the Balkans is characterized not by a lack of awareness, but by limited structural integration between nanotechnology and plant biotechnology research communities. Opportunities to apply this knowledge are often constrained by fragmented institutional structures, restricted interdisciplinary funding, and limited access to shared nanomaterial characterization facilities. As a result, the existing expertise and awareness have not yet translated into tangible collaborative projects or peer-reviewed outputs in the field of woody plant in vitro biotechnology. Even for herbaceous species, nano-assisted in vitro studies are scarce, with only a few documented trials reported by a research group in Bulgaria for *Stevia rebaudiana* Bertoni [31,32,33,34]. This lack of regional activities in the field of nano-assisted in vitro systems defines a clear “nano gap” in the Balkans, which should become a priority for both basic and applied research to make progress in this under-researched area.

While nanotechnology can also play a role in genetic transformation and nanoparticle-mediated gene delivery, these topics fall outside the scope of this review. The focus here is specifically on in vitro micropropagation and preservation systems for woody plants, where nanotechnology applications remain largely unexplored.

The aim of this paper is therefore to assess international progress in nanotechnological applications for in vitro propagation and conservation, evaluate the status and capacities in the Balkans and identify strategic, low-threshold entry points for implementation, with a focus on woody species. Unlike previous global reviews, this paper provides the first region-specific assessment linking nanotechnology capacities with in vitro biotechnology in woody plants. It identifies the unaddressed “nano gap” in the Balkans, formulates a framework for its reduction, and proposes actionable regional strategies. Thus, the paper represents a new analytical contribution that connects scientific advancement with policy and cooperation dimensions. By applying this framework to the Balkan context, the review provides insights into how structural and interdisciplinary factors influence the translation of nanotechnology into applied in vitro systems. This perspective makes the study relevant not only regionally, but also for understanding how emerging scientific regions can align with global innovation trends.

## 2. Applications of Nanotechnology in Plant Tissue Culture: Opportunities for the Balkan Countries

Nanotechnology has proven to be a promising avenue in plant tissue culture, offering innovative solutions that facilitate stress reduction, improve micropropagation rates, increase cryopreservation efficiency, and promote the synthesis of secondary metabolites. Conventional in vitro methods have evolved significantly in the Balkans; however, nano-assisted methods are not yet proven in the region. This section describes the potential applications of nanomaterials for woody plants, referring to the global developments and highlighting opportunities for their integration into in vitro plant biotechnology in the Balkans.

### 2.1. Nano-Assisted Surface Sterilization

Microbial contamination is a constant problem in all tissue culture laboratories, especially when working with explants derived from field-grown or wild plants [35,36]. Various studies have been conducted in the Balkans to determine the most effective methods for sterilizing horticultural woody plants, including berry fruits, stone and pome fruits, nuts and grapes, as well as various forest species [1]. Sterilization of the explants is a crucial step, regardless of the purpose of the in vitro culture; therefore, considerable efforts are made to ensure that this step is carried out correctly and that the experimental platform can continue to run. Nevertheless, it must be noted that in the Balkans there are no reports on the use of nanotechnology to stabilize aseptic cultures, not only in woody plants but also in herbaceous plants.

Several studies outside the Balkans have shown that nanoparticles can significantly reduce microbial contamination [37]. Most reports focus on herbaceous plants, but an increasing number of studies show that woody horticultural and forest plants can be utilized in various ways in the future. For example, Gouran et al. [38] found that AgNPs at concentrations between 100 and 1000 ppm were effective in reducing microbial contamination of grapevines. Arab et al. [39] observed similar effects when using AgNPs on almond × peach rootstock at concentrations of 100 and 150 ppm. El-Kosary et al. and El-Sharabasy and Zayed [40,41] found that the addition of AgNPs to the culture medium facilitated the stabilization of aseptic cultures in date palms. Other reports, such as for *Olea* sp. [42] and *Capparis* sp. [43], have shown that AgNPs can be used to mitigate contamination. In addition, Zakhalova et al. [23] used titanium trisulfide nanoribbons to improve the efficacy of sterilization and the success of micropropagation in poplar × aspen hybrid explants. In a separate study, Zakhalova et al. [44] used AgNPs to reduce the contamination rate in hairy birch explants.

### 2.2. Nano-Enhanced Micropropagation

Many successful reports from the Balkans show that considerable efforts have been made to increase the micropropagation rates of woody plants. Researchers have conducted extensive studies on different woody species to improve their growth and rooting to ensure their mass production through micropropagation in semi-solid or temporary immersion system (TIS) bioreactors [1]. Although these are significant advances, there are no reports of nano-enabled applications to improve the micropropagation stages of species in the Balkan region.

Nanoparticles have been applied to various woody plant species in different parts of the world and have been shown to increase yield in plant biotechnology. For example, ZnO NPs at concentrations of 50 and 75 mg/L have been reported to improve shoot regeneration from callus and root induction in date palms [45]. ZnO NPs at a concentration as low as 5 mg/L significantly increase the amount of biomass produced by pomegranates [46]. Furthermore, both ZnO NPs and CuO NPs have been shown to effectively promote the in vitro development of *Citrus* sp. [47]. There is significant research data on the use of nanoparticles to enhance micropropagation of olive (*Olea europaea*). In this case, Ag NP at a concentration of 20 and 30 mg/L was found to accelerate shoot growth [48], while ZnO-NP at a concentration of 6 and 18 mg/L was found to stimulate in vitro regeneration [49]. It was also shown that AgNP at a very low concentration of 1.5 μg/L induced the growth of roots in gray poplar (*Populus* × *canescens*) [50].

The efficacy of ZnO NPs and Ag NPs in promoting in vitro regeneration of *Betula pubescens* and a hybrid of *Populus pyramidalis* × *Populus nigra* is very well documented [51]. Another study reports on the efficacy of CuO NPs in the in vitro culture of *Populus alba* × *Populus tremula* [52]. When comparing the cultivation of *Prunus armeniaca* on semi-solid and TISs in the presence of Ag NPs, it was found that cultivation in TIS in the presence of Ag NPs at a concentration of 100 mg/L was the best treatment for in vitro regeneration [53]. On the other hand, the addition of silica-based nanoparticles to the regeneration medium significantly increased the number of lateral shoots in Myrobalan 29C (*Prunus cerasifera* L.) [54]. Carbon nanotubes have also been shown to be effective in increasing in vitro production. Single-walled carbon nanotubes (SWCNTs) were found to enhance all micropropagation stages of blackberry plantlets [21].

The researchers also investigated the efficacy of different types of nanoparticles in comparison to each other [17,18]. When comparing the effects of ZnO NPs, iron oxide (Fe_2_O_3_ NPs), aluminum oxide (Al_2_O_3_ NPs), and titanium dioxide (TiO_2_ NPs) on the micropropagation of *Sequoia sempervirens*, it was found that TiO_2_ NPs at a concentration of 10 mg/L were most effective in promoting plant development [17]. When investigating the use of metallic nanoparticles, in particular Fe_2_O_3_ NPs, ZnO NPs, and Mn_2_O_3_ NPs, in the in vitro culture of *Populus alba*, it was concluded that ZnO NPs had the most favorable effect on growth rates [18].

The use of nano-chelated forms of zinc, iron, or boron to improve nutrient uptake during the critical early stages of shoot and root induction could prove effective for woody plant cultures, as many recalcitrant genotypes exhibit poor in vitro growth due to micronutrient imbalances or limitations in uptake. In this context, the above findings can be utilized to develop applications for micropropagation of autochthonous Balkan plant species.

In addition to enhancing morphogenesis and growth, nanoparticles may also influence the genetic stability of regenerants. Their presence in in vitro culture can affect normal cellular processes and lead to genetic and epigenetic changes in regenerants, potentially promoting heritable variability. While acute responses in in vitro plants may manifest as immediate toxicity, low doses applied could lead to discreet but cumulative physiological and epigenetic shifts during repeated subculturing [55]. Prasad et al. [15] reviewed the broader role of metallic NPs in organogenesis and morphogenesis, suggesting that NP-induced oxidative stress or alternation in chromatin structure can induce genomic variation that sometimes could be beneficial for breeding programs seeking novel traits. However, they have to be applied cautiously to avoid undesirable genomic instability. Studies in several herbaceous species have reported an increase in ploidy levels, changes in methylation status, modifications in protein and DNA content, as well as anatomical alterations when calluses or regenerating tissue were cultured with carbon, silver, or gold particles [12,30]. However, experimental studies directly examining NP-induced somaclonal variation in woody plant species remain scarce; most of the available evidences come from herbaceous systems. Research on the influence of AgNPs on the in vitro growth of olive (*Olea europea*) revealed that these nanoparticles affect leaf anatomy, and the response was genotype- and concentration-dependent [48]. The authors reported morphological and anatomical changes that may be associated with somaclonal variation and stress responses.

### 2.3. Enhancing Survival and Delaying Explant Senescence

Recent studies also suggest that nanomaterials can play a significant role in enhancing explant survival and extending explant lifespan in woody plant tissue culture [13]. Findings by Sarmast et al. [56] indicate that low concentrations of AgNPs delay the onset of senescence in *Tecomella undulata* (Roxb.) Seem. tissue culture by reducing ethylene biosynthesis. In media containing AgNPs, even at the lowest concentrations, *TuACS* expression was reduced, which could partly explain the delayed senescence of explants and their improved survival under in vitro conditions. However, concentrations above 60 mg L^−1^ were reported to be detrimental to shoot regeneration. These effects may also involve hormonal cross-talk among ethylene, auxin, gibberellin, and jasmonic acid pathways. In summary, the authors suggest that AgNPs might be suitable for use in crops with yields affected by ethylene sensitivity.

### 2.4. Nano-Encapsulation for the Controlled Release of Bioactive Substances

In the Balkans, several efforts have been made to optimize the composition and absorption of culture media. However, applications involving nano-encapsulation techniques to regulate the release of certain chemicals have not yet been applied in the region. Even globally, few studies have been conducted on this topic that could help in cloning woody plants in the Balkans. Nanocapsules containing rhizobacteria have a positive effect on the in vitro development of pistachios [57]. In relation to herbaceous plants, a study on *N. tabacum* [58] showed that auxin-stabilized Ag NPs significantly enhanced rhizogenic responses in plants grown in vitro. Nano-encapsulated plant growth regulators could effectively increase agricultural productivity.

In agricultural practice to increase crop production, these techniques appear to be beneficial and well accepted [59,60] and thus represent a valuable initiative that can be considered to advance in vitro research and applications in a variety of plant species.

### 2.5. Nano-Assisted Cryopreservation

Cryopreservation in the Balkans is already well advanced in Serbia, while other countries are still in the early stages or have not yet started long-term storage of woody plants [1]. Even in this case, nanotechnology has not yet been integrated into the cryogenic conservation protocols applied in the region so far. It is important to mention that there are only a limited number of studies in this field worldwide on both woody and herbaceous plant species. Among the few available examples is the study in which Ag NP was effectively used in the preservation solution for the deep-freeze storage of embryogenic cell lines of *Pinus radiate*, achieving a high regeneration rate [61]. In *Lamprocapnos spectabilis*, an herbaceous plant, the effectiveness of gold nanoparticles (Au NPs) in cryopreservation has reported [62]. In addition, SWCN increased the survival rate of *Agapanthus praecox* [63]. Although such applications still remain limited worldwide, they have great potential to be extended in many plant species and thus establish effective protocols for long-term conservation protocols.

### 2.6. Nano-Enhanced Production of Secondary Metabolites

Research on the production of secondary metabolites from woody plants in the Balkan region has so far been pursued in Serbia [1]. However, there are currently no reports of in vitro nano-mediated improvements in this area for woody plant species. In contrast, studies from abroad have shown the significant potential of nanotechnology in stimulating the biosynthesis of secondary metabolites. For example, the enhanced synthesis of the anticancer drug paclitaxel by adding calixarene (CAL)-based nanoparticles to the culture media of *Taxus* sp. cell cultures has been reported [64]. Similarly, the addition of ZnO NPs to cultures of *Delonix* spp. increased the accumulation of bioactive secondary metabolites [65]. The application of Ag NPs at the concentration of five ppm in suspension cultures of *Corylus* spp. resulted in a significantly higher yield of taxanes compared to untreated controls [66]. On the other hand, the addition of 20% selenium nanoparticles (Se NPs) to the culture medium stimulated a higher production capacity of bioactive compounds in callus cultures of *Populus* sp. [67].

In the Balkans, there are three remarkable reports from Bulgaria on the improvement of secondary metabolism of herbaceous plants by the use of nanoparticles [32,33,34]. Their studies reported that the use of silver salt nanofibers stimulated the synthesis of steviol glycoside content in the micropropagated plants of *Stevia rebaudiana* Bert., and thus provided a relevant starting point for the extension of such applications to woody plants in the region [32,33]. A recent study has shown that organic nanofibers promote the production of specific metabolites and the antioxidant potential of *Stevia rebaudiana* in vitro cultures [34].

### 2.7. Nano-Assisted Stress Attenuation

The use of in vitro culture systems to reduce the effects of abiotic stress and select stress-tolerant plant varieties is of increasing importance, especially in the context of accelerating climate change. In the Balkan region, Serbia is a leader in in vitro stress research on woody plants; however, in this context, no research has yet been conducted on the inclusion of nanotechnology in stress reduction studies. Conversely, research from other countries has shown the effectiveness of nanoparticles in alleviating stress symptoms in various woody plant species. The addition of Fe_2_O_3_ NPs at concentrations of 5 or 10 mg/L to the culture medium significantly alleviated salt stress in *Populus* sp. [68]. Iron nanoparticles reduced the negative effects of salt stress in grapevines [69]. In *Phoenix dactylifera* (date palm), the addition of 1 mg/L iron nanoparticles to the in vitro growth medium significantly reduced both drought and salt stress [70].

An example of nano-assisted stress reduction in herbaceous species under in vitro conditions in the Balkans also comes from Bulgaria. Miladinova-Georgieva et al. [31] reported that pretreatment of *Stevia rebaudiana* plantlets with silver-containing amino acid nanofibers improved ex vitro adaptation and increased drought tolerance.

Although the application is limited to woody taxa, these positive effects of nano applications to improve micropropagation open the door to promote in vitro organogenesis in recalcitrant species (Figure 1). Institutions and research groups in the Balkan region dealing with indigenous woody horticultural and forestry species could benefit from the reported studies and start practical applications for their autochthonous germplasm. Pilot studies in regional tissue culture laboratories, particularly in Serbia, Bulgaria, Croatia, Albania and Greece, could focus on the application of low-dose nanoparticle treatments aimed at modulating hormonal responses and improving the in vitro production and preservation of woody plant species.

## 3. Cross-Sector Applications of Nanotechnology: A Regional Starting Point

Although nanotechnology has not yet been applied to in vitro culture systems for woody species in the Balkans, the region is not starting from zero. Several research institutions in the Balkans where plant tissue culture techniques are applied have developed extensive expertise in the nanoparticle synthesis, characterization and application of nanoparticles. This section focuses on advances, particularly in the environmental and agricultural fields, supported by trained professionals and researchers who ensure both technical capacity and scientific expertise. These foundations provide an essential opportunity to translate existing knowledge into future applications of in vitro plant biotechnology. Figure 2 illustrates the comparative status of in vitro woody plant biotechnology in nine Balkan countries, assessing nanotechnology capacities in general areas and actual nano-integration into in vitro systems. This assessment is made only for countries where the plant tissue infrastructure is currently functional.

**Albania** has relatively modest experience with nanotechnological applications, particularly in the fields of plant and environmental sciences. Some progress has been made by researchers at the Agricultural University of Tirana in studies in the field of agriculture. In this context, Prifti and Maçi [71,72] demonstrated the positive effect of herbagreen nanoparticles on increasing the productivity of cereals. Researchers from the Faculty of Natural Sciences (FNS) at the University of Tirana are also contributing to this topic. One study focused on the structural characterization of natural clay from south-eastern Albania [73], which is essential for various applications, including horticulture and plant growth. In addition, the study by Dervishi et al. [74] investigated the photocatalytic properties of a novel Ag/TiO_2_/graphite nanocomposite, which is widely used in pollution control and agriculture due to its antibacterial and antifungal properties. In parallel, the Department of Biotechnology (FNS, University of Tirana) has made remarkable progress in the further development of in vitro techniques for the propagation and conservation of plant germplasm. The fact that both biotechnological and nanotechnological research are located at the same faculty creates a favorable environment for advancing multidisciplinary studies in the future. An important milestone was the establishment of NanoAlb (https://nanoalb.al/nanoalb/index.php; accessed on 4 September 2025) by the Academy of Sciences of Albania in November 2019. This initiative was developed to support, coordinate, and harmonize scientific research and interdisciplinary collaboration between Albanian research groups across institutions and disciplines. Research groups from Kosovo and North Macedonia were also part of this scientific unit. In 2024, this initiative was further expanded with the transformation of NanoAlb into NanoBalkan, a regional research center for nanotechnology. The focus is on promoting cross-border cooperation within the Balkan region and beyond (https://www.nanobalkanconf.com/2025/; accessed on 4 September 2025) and on potential advances through interdisciplinary cooperation in in vitro plant biotechnology.

In **Bosnia and Herzegovina**, there are currently no nano-applications in in vitro systems for either woody or herbaceous plant species, indicating a significant gap and an area with high development potential. Although there are currently no direct applications in in vitro systems, related fields such as agriculture and environmental protection are increasingly turning to modern technologies. At the University of Banja Luka, research related to the application of nanotechnology is primarily focused on environmental protection and chemical engineering. For example, one chapter of the monograph “The Environment” reports on the application of nanotechnology in environmental protection, including the catalytic degradation of pollutants, adsorption and filtration, and the production of nanomaterials from industrial waste [75]. In addition, Gotovac Atlagić et al. [76] pointed out the potential of valorization of industrial waste for the development of functional nanomaterials. The thermal and structural stability of natural zeolite was analyzed, with the results indicating its suitability for applications in adsorption, catalysis, and the development of nanotechnological processes in environmental protection [77]. The concept of ‘smart agriculture’ is gradually being introduced with the help of IoT solutions, sensors, automation, drones and digital platforms and is supported by the European Union and academic institutions. In addition, research institutions such as the Institute for Genetic Engineering and Biotechnology (INGEB) in Sarajevo, the Institute for Genetic Resources (IGR) in Banja Luka, and the Faculty of Agriculture of the University of Banja Luka (UNIBL) are actively involved in the development of in vitro plant models and the conservation of plant genetic resources. These institutions are promoting the potential in the field of biotechnology and are working on in vitro models for testing bioactive substances and food safety, which can serve as a basis for the future integration of nanotechnology in plant systems in agriculture, horticulture, and forestry. Although there are currently no direct nano-applications in in vitro cultures in Bosnia and Herzegovina, the existing knowledge, institutional infrastructure, and regional commitment to modern technologies in agriculture, horticulture, forestry, and environmental protection—together with the efforts of key scientific institutions—indicate favorable conditions for the development of such applications in the near-future.

**Bulgaria** has recently presented several important studies on the application of nanotechnology in agricultural sciences. Krumova et al. [78] found in their study that chitosan-based nanocomposites showed superior antifungal efficacy, with CS-ZnO NCs being the most effective. In addition, both chitosan and ZnO NPs showed pronounced antifungal activity against the two pathogenic strains *Fusarium solani* (*F. solani*) and *Alternaria solani* (*A. solani*) and effectively inhibited mycelial growth [79]. The authors propose chitosan and zinc oxide nanoparticles as potential fungicides against *A. solani* and *F. solani*. A similar antifungal effect of metal and metal oxide nanoparticles was demonstrated on *Verticillium dahliae* [79]. The studies with *Stevia rebaudiana* Bertoni tissue cultures showed that the controlled application of nanofibers could be a valuable strategy to optimize Stevia propagation and metabolite production [31,32,33,34].

**Croatia** currently lacks systematic investigations and peer-reviewed publications on the application of nanotechnology in the plant tissue culture of woody species, but emerging possibilities point to a growing and promising area of research. One particularly important contribution is a PhD thesis dealing with the application of SiO_2_ and ZnO nanoparticles to optimize micropropagation of blueberries (*Vaccinium corymbosum* L.) in modern temporary immersion bioreactor systems [80]. This research work is in the final stage of preparation for a scientific publication which is expected to appear soon and will represent the first Croatian scientific contribution in the field of integration of nanotechnology and in vitro plant tissue culture. In addition, the repository of the Faculty of Agrobiotechnical Sciences Osijek (FAZOS)—University of Osijek (UNIOS), contains several bachelor and master’s theses dealing with topics related to nanotechnology in the plant tissue culture of woody plants, which confirms the growing interest of young researchers. In Croatia, scientific research into the application of nanotechnology is primarily focused on the fields of medicine and toxicology. For example, the project “Nanosilver Phytotoxicity: Mechanisms of Action and Interaction in Tobacco Cells (NanoPhytoTox)”, which was funded by the Croatian Science Fondation, led to the publication of several scientific papers [81,82,83,84] which investigated the mechanisms underlying the phytotoxic effects of AgNPs in tobacco plants (*Nicotiana tabacum*). The research focused on different types of AgNPs stabilized with different surface coatings (citrate, polyvinylpyrrolidone—PVP, and cetyltrimethylammonium bromide—CTAB) and ionic silver (AgNO_3_). Key aspects included the stability of the nanoparticles in the exposure medium, uptake and accumulation in plant tissue, and their effects on photosynthesis, pigment content, chloroplast ultrastructure, and response to oxidative stress. Both seedlings and adult plants were analyzed to assess the sensitivity to AgNPs depending on the developmental stage. In summary, the NanoPhytoTox project has shown that the toxicity of silver nanoparticles to plants is strongly influenced by their physicochemical properties (size, stability, surface coating, and charge) in addition to the release of ionic silver. These results emphasize the need to consider nanoparticle-specific properties when assessing environmental safety and provide a scientific basis for safer applications of nanotechnology in agriculture. Croatia is at an early stage of integrating nanotechnology into plant tissue culture, but initial contributions indicate clear potential. Interdisciplinary and regional co-operation will be key to closing research gaps and promoting innovative solutions for sustainable agriculture and forestry.

In **Greece**, there is no evidence in the current literature of the use of nanotechnology in the micropropagation of plants. On the other hand, all mentioned studies explore the use of various nanoparticles (NPs) in plant science/agriculture, primarily for the plant protection or plant health/yield improvement. Nanotechnology in crop protection is primarily focused on the development of nano-sized fungicides and bactericides. These are intended to be more effective, environmentally friendly, and resistance-breaking alternatives to conventional pesticides by having direct antimicrobial effects or inducing plant defense. In particular, Malandrakis et al. [85] used NPs of copper (Cu NPs, CuO NPs), silver (Ag NPs) and zinc (ZnO NPs) to control plant pathogens in palm trees. Similarly, copper NPs have been shown to be effective in controlling bacterial spot disease in tomatoes, particularly against *Xanthomonas campestris* pv. *vesicatoria*, often outperforming conventional copper formulations [86]. In addition, novel copper-based NPs were tested and their activity was successfully evaluated both in vitro and in planta against important olive pathogens such as *Fusicladium oleagineum* (olive leaf spot disease) and *Colletotrichum* spp. (anthracnose) [87]. Beyond the direct antimicrobial effect, some nanoparticles, such as jasmonic acid loaded chitosan NPs (JA-CNPs), have been shown to reduce the severity of *Botrytis cinerea* infection in *Arabidopsis thaliana* and tomato leaves by stimulating the plant’s own defense mechanism [88]. In addition to their role in direct plant protection, NPs are also being investigated for their influence on plant physiology, growth, and even complex biological interactions. The effects of copper (Cu NPs, CuO NPs), silver (Ag NPs), and zinc oxide NPs (ZnO NPs) on the growth of tomato plants, their physiological properties, and their symbiotic relationship with the endophytic strain *Fusarium solani* FsK are well documented [89]. The age-dependent effects of foliar-sprayed copper-zinc NPs (CuZn) on photosynthetic efficiency and reactive oxygen species (ROS) formation in *Arabidopsis thaliana* leaves have shown different responses between young and mature leaves [90]. Further studies on photosynthetic function include work by [91], who synthesized and evaluated pegylated zinc-doped ferrite NPs (ZnFer) and investigated their effects on PSII function in tomato plants following foliar application. The importance of nanoparticle morphology and coating for these interactions was emphasized by Tryfon et al. (2023) [92], whose study of rod-shaped oleylamine-coated zinc oxide nanorods (ZnO@OAm NRs) in tomato plants showed significant effects on photosynthetic function and ROS formation compared to irregularly shaped ZnO nanoparticles. It can be concluded that research on the application of nanotechnology in agriculture in Greece shows a strong drive to exploit the unique properties of nanomaterials to improve agricultural sustainability through enhanced disease control and optimization of crop performance, while basic research is being conducted to understand the complex and often nuanced interactions between nanoparticles and plant biology.

**North Macedonia** is still in the early stages of research in the field of nanotechnology for agricultural and ecological applications and is actively developing further. The current focus is on innovative solutions to local challenges, including water purification, soil remediation, and the development of “smart” agricultural inputs (nano-agrochemicals). So far, work has centered on the publication of review reports [93,94], which have helped to accumulate theoretical knowledge in the field. This may contribute to future advances in nano-applications in the country.

In **Serbia**, several research groups are investigating the interactions between plants and nanoparticles and the application of nanotechnology in agriculture. Most studies focus on “plant nanobionics”, which aims to improve the physiological functions of plants, although there is still some pioneering work on woody species. Recently, natural calcite nanofertilizer (Fitosmart^®^) was tested on grapevines, which showed an improvement in yield, quality and biochemical composition of the fruit [95]. In addition to woody plants, there is also a lot of data on the application of nanoparticles to various herbaceous species in Serbia. For example, foliar application of orange carbon dots (oCDs) in green beans and maize, including under field conditions, has been extensively studied and patented with results demonstrating their great potential as biostimulants [96,97,98,99]. The same group also investigated polymer-coated CeO_2_ nanoparticles in wheat and peas and analyzed the effects on germination, growth, and metabolism [100,101,102]. DNA-coated tungsten nanoparticles carrying the GSH1 gene (γ-glutamylcysteine synthetase) were used for biolistic transformation of mature wheat embryos and stable integration into the wheat genome to improve its tolerance to drought stress [103]. While genetic transformation lies beyond the focus of this review, such examples highlight the advanced nanobiotechnological expertise present in the region, which could facilitate future interdisciplinary integration with in vitro plant biotechnology. Another team focused on fullerenol nanoparticles and their effects on water metabolism and drought resistance of sugar beet and *Arabidopsis thaliana*, combining physiological, biochemical, and molecular approaches [104,105]. Fullerol nanoparticles affect the growth of wheat under drought stress and evaluate fungal (*Aspergilus flavus*) aggressiveness, secondary metabolites, and mycotoxin contamination during storage [106]. Other contributions include nanopriming of seeds with different nanomaterials to improve germination, early growth, and drought tolerance of different plant species such as wheat [107], tomato [108], pea [109,110] and radish [111]. Recently, the effect of Fitosmart^®^ nanofertilizer on the growth of strawberries under greenhouse conditions was documented [112]. There are several national and international projects supporting this area. The project “Nanobionic stimulation of agricultural plants’ productivity” (2020–2021, Innovation Fund) resulted in a patent on the use of organic nanoparticles to increase plant productivity (https://rimsi.imsi.bg.ac.rs/handle/123456789/1854; accessed on 4 September 2025). Currently (2025–2027), an active bilateral project “Utilizing nanotechnology and sustainable approaches for environmental remediation in the agri-food systems” is underway between the Faculty of Science of the University of Novi Sad (Serbia) and the University of Torino (Italy) which deals with the application of nanotechnology for environmental remediation (https://www.pmf.uns.ac.rs/en/?s=Utilizing+nanotechnology+and+sustainable+approaches+for+environmental+remediation+in+the+agri-food+systems; accessed on 4 September 2025). The EU-funded FORTILEAVES project (2024–2026) is developing biofortification technologies that combine microorganisms and nanotechnology (https://hungary-serbia.eu/projects/fertileaves; on 4 September 2025). The BioSense Institute (University of Novi Sad) is participating in the Horizon Europe project Know4Nano (2024–2027), which is developing biosensors for the detection of pathogens [113]. The AerPotato project (2024–2026, Institute for Biological Research “Siniša Stanković”), which is funded by the National Science Fond of Serbia, is researching nanoparticles to improve seed potato yields in aeroponic systems at elevated temperatures.

These fundamental efforts underline the technical readiness of the region and the current level of expertise in the field of agriculture and environmental nano-applications. More importantly, they provide a strategic opportunity to transfer existing capabilities to the field of in vitro biotechnology for woody plants. Figure 2 illustrates the “nano gap” in the third column and shows that several countries have medium to high competences in the first two areas, but have not yet translated them into nano-enabled applications in woody plant tissue culture. This figure provides a qualitative overview intended to illustrate the relative advancement of traditional in vitro systems and the absence of nano-assisted applications. Due to the lack of published studies on nanointegration in the region, the comparison does not include quantitative data. Strengthening this interdisciplinary integration is essential for closing the current “nano gap” and aligning regional research with broader European innovation priorities in sustainable agriculture and biotechnology.

## 4. Reasons for the “Nano Gap” and Strategic Ways to Integrate into in Biotechnology for Woody Plants

Through a comparative assessment of global nano-applications and their absence in in vitro systems in the Balkans, the analysis identifies key patterns and preconditions for successful technology integration. The identification of these enabling factors allows the review to function as both a diagnostic and strategic tool for guiding regional capacity building and innovation policy.

A clear *lack of collaboration* persists between nanoscience researchers and plant biotechnologists. However, this disciplinary separation represents only one dimension of the broader “nano gap” At a deeper level, the gap reflects *systemic socio-economic and infrastructural imbalances* across the Balkan region. Limited public and private investment in R&D, the absence of dedicated national programs for nanobiotechnology, and fragmented institutional frameworks reduce the capacity to sustain interdisciplinary innovation. Many laboratories lack access to high-precision analytical equipment and shared nanomaterial synthesis facilities, creating a technological barrier to entry into nanobiotechnology research. The limited number of nano-assisted studies should not be interpreted as a lack of scientific awareness but rather as a reflection of the absence of enabling frameworks and incentives that support cross-sectoral experimentation and innovation.

Furthermore, *policy and governance factors* play a significant role. National science strategies often treat nanotechnology and biotechnology as distinct thematic areas, with minimal overlap in funding calls or evaluation panels. Regional disparities in research infrastructure are compounded by unequal participation in EU Horizon and COST frameworks, making cross-border initiatives more difficult to implement. In addition, limited industrial involvement and weak technology-transfer ecosystems reduce the demand for applied nanobiotechnological solutions, further slowing progress.

These structural, economic, and policy-related combined factors define the multidimensional nature of the “nano gap,” illustrating that bridging it requires coordinated strategies that extend beyond academic collaboration to include innovation policy, regional funding mechanisms, and capacity building at multiple levels.

Another set of obstacles concerns *research fragmentation and methodological limitations*. Scientific projects are often narrowly focused and lack interdisciplinary initiatives that bring together nanoparticle synthesis, plant tissue culture, and in vitro physiology. In addition, limited research funding hampers progress in nanotechnology, which depends on sophisticated equipment and infrastructure.

Although the European Union has established a regulatory framework for nanotechnology, its implementation at the regional level remains challenging, particularly with regard to novel agricultural applications [114,115]. The absence of standardized experimental protocols for evaluating nanoparticle behavior in plant tissue culture systems further slows integration, as results from different studies are difficult to compare or reproduce [116]. Moreover, ongoing uncertainties about nanoparticle accumulation and long-term ecotoxicological effects contribute to cautious adoption [117].

To bridge the current gap and build on existing regional capacities, research groups from different fields should consider several steps that may be helpful in this regard (Figure 3):Organizing regional workshops and training programs is considered a valuable way to bring together experts from the fields of nanotechnology, agriculture, and in vitro plant biotechnology. Such activities promote knowledge exchange and collaboration in the development of multidisciplinary experimental platforms and contribute to building practical regional expertise. Good initiatives so far are the international activities organized by NanoBalkan, which have become a tradition in Albania for several years (https://www.nanobalkanconf.com/2025/; accessed on 4 September 2025). The participation of a larger number of scientists from different Balkan countries working in the field of in vitro plant biotechnology and agriculture could create good opportunities for multidisciplinary integration with nanosciences.Assess the current state of plant tissue culture and nanobiotechnology curricula in the region’s universities and endeavor to integrate them in the best possible way in order to train and better prepare the new generation of workers in this field.Seek effective funding sources and implement multidisciplinary projects to support nanobiotech studies, shared infrastructure, and applications in high-priority species. These funds could be used in pilot projects to optimize protocols, make an initial assessment of results and raise awareness among researchers of the benefits of multidisciplinary collaboration.International networks, such as CopyTree (https://www.cost.eu/actions/CA21157/; accessed on 4 September 2025), CryoConnect (https://www.ecpgr.org/working-groups/cryopreservation/cryoconnect; accessed on 4 September 2025) and the NanoBalkan Research Center, could help in integrating nano-research into existing international initiatives for the propagation and conservation of woody plants. Regional consortia can be useful for sharing best practices and collaborating on joint projects involving nano-applications in plant tissue culture, large-scale plant production, and the development of elite cultivars.A proactive approach to the EU regulatory framework is extremely important, as it helps to ensure the safe development and application of nanotechnologies in agriculture and in all plant sciences. In this context, translating regulations into practical guidelines for regional researchers and companies can be beneficial for future advances in nanotechnology.The establishment of joint research facilities and open-access nanotech laboratories could be another approach to close the existing gap. Such facilities would enable the sharing of resources, reduce costs and provide equal access to advanced equipment and knowledge, which would facilitate the application of nanotechnologies in plant sciences.

## 5. Conclusions

This review highlights the presence of a distinct “nano gap” in the integration of nanotechnology into in vitro biotechnology for woody plants in the Balkan region. Despite significant regional progress in the propagation, conservation, and stress mitigation of woody species through conventional in vitro approaches, nanotechnology has not yet been integrated into these systems, and there are no officially published results. In contrast, global research indicates that nanoparticles and nanomaterials can significantly enhance the success of in vitro aseptic culture establishment and stabilization, propagation efficiency, cryopreservation results, and secondary metabolite production, thereby improving the reliability and robustness of in vitro protocols.

The limited adoption of nanoparticle-based applications in the Balkans appears to result less from technical incapacity than from broader structural, economic, and organizational barriers, including fragmented research efforts, insufficient cross-disciplinary collaboration, underfunding, and the absence of standardized protocols. Nevertheless, the region possesses significant expertise in nanoscience, agriculture, and biotechnology, indicating that conditions are favorable for advancing research in these areas.

Coordinated strategies are needed to bridge this gap, including the establishment of multidisciplinary consortia, the development of co-operative research infrastructures, the integration of nanobiotechnology into university curricula, and the initiation of pilot projects for woody plant species. In addition, alignment with EU research initiatives and regulatory frameworks is essential to achieve both safety and broader scientific impact.

From a broader perspective, nanotechnology offers major opportunities and challenges. Nanomaterials can help propagate endangered species, boost crop yields and stress tolerance, and enable the controlled delivery of agrochemicals and bioactive compounds, thereby supporting more sustainable plant production systems. At the same time, concerns about nanoparticle uptake, movement within plants, and environmental accumulation highlight the need for standardized methods and clear regulations to ensure that plant nanobiotechnology develops safely and responsibly.

Beyond summarizing current knowledge, this study introduces the concept of a “nano gap” as a practical tool for understanding the distance between nanoscientific potential and its real-world application in plant biotechnology. The proposed roadmap combines technical, educational, and policy-focused strategies to help close this gap and promote sustainable, inclusive regional development.

## Figures and Tables

**Figure 1 plants-14-03499-f001:**
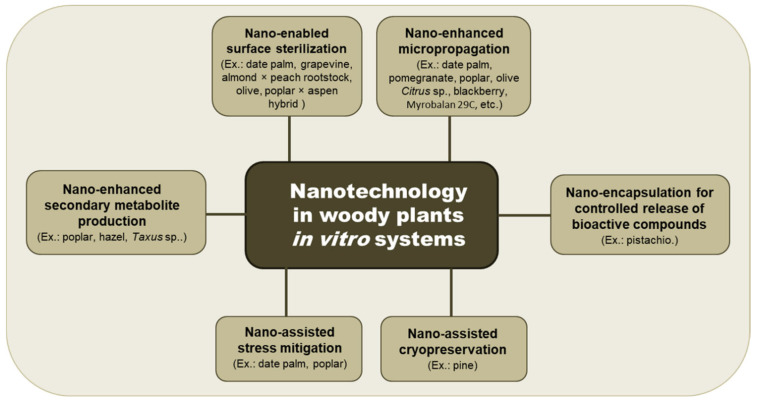
Schematic summary of nano-applications in the in vitro biotechnology of woody plants.

**Figure 2 plants-14-03499-f002:**
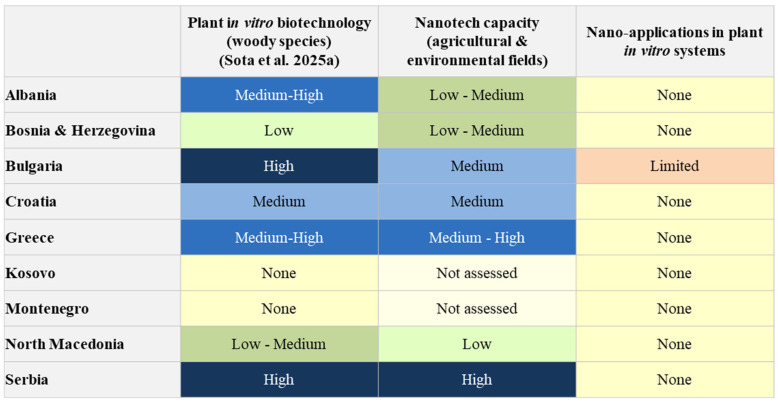
Conceptual comparison between the level of development of in vitro plant biotechnology and the current status of nanotechnology integration in the Balkan region [1].

**Figure 3 plants-14-03499-f003:**
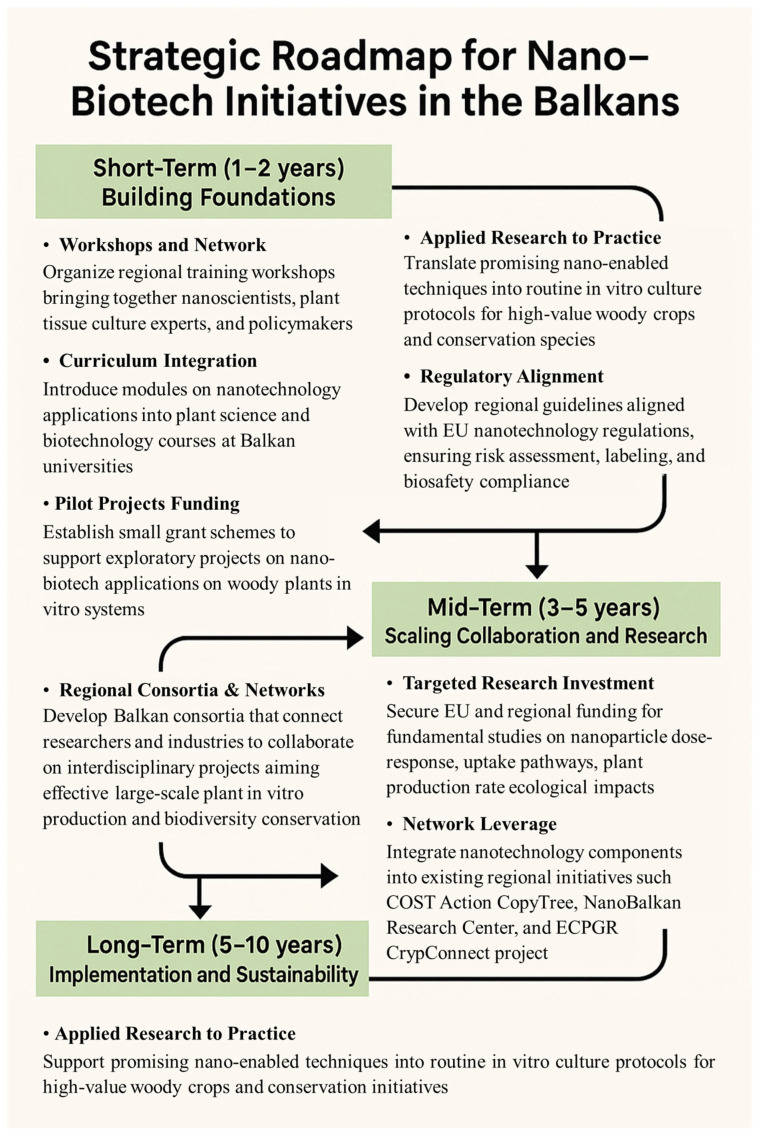
Strategy for the integration of nanotechnology into woody plant in vitro biotechnology in the Balkans.

## Data Availability

No new data were created or analyzed in this study. Data sharing is not applicable to this article.

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
