# Peer review of "The Balkan Region and the “Nano Gap”: An Underexplored Dimension of In Vitro Biotechnology for Woody Plants"

_plants, 2025, doi:10.3390/plants14223499_

Round 1
Reviewer 1 Report
Comments and Suggestions for Authors
This review highlights the existence of a clear "nano-gap" in the integration of nanotechnology into in vitro biotechnology for woody plants in the Balkans.
The article aims to assess the current situation and capacities in the Balkans regarding nanotechnological applications for in vitro propagation, surface sterilization, conservation, production of secondary metabolites, nano-encapsulation and to define the strategic landscape, focusing on woody species. This approach will facilitate alignment with broader European strategies for sustainable agriculture, biodiversity conservation, and green innovation, thereby strengthening collaboration between academic institutions and industry stakeholders.
The manuscript highlights numerous areas related to the use of nanoparticles and their status in the Balkan countries. It also highlights the cross-sectoral applications of nanotechnology, highlighting the existence of a clear "nano-gap" in the integration of nanotechnology into in vitro biotechnology.
Therefore, the manuscript contains important and valuable information and should be accepted as is.
Best Regards,
Author Response
Dear Reviewer,
We thank you for your positive comments, which highly motivate us to continue our work in the field.
Kind regards,
Corresponding authors
(on behalf of all authors)
Reviewer 2 Report
Comments and Suggestions for Authors
The review article entitled “The Balkan region and the Nano Gap: An underexplored dimension of in vitro biotechnology for woody plants” is generally well-written and fits within the scope of this journal. The manuscript discusses the limited utilization of nanotechnology in woody plant tissue culture within the Balkan region, in comparison to global advancements. However, the manuscript has several major issues that need to be addressed before it can be considered for publication.
Major Comments:
-
Novelty: The authors should clearly articulate the novelty and scientific contribution of this review. As it currently stands, the manuscript primarily summarizes existing research and does not highlight any new perspectives or critical insights.
-
Scope and Focus: The review mainly emphasizes that the Balkan region has not yet adopted nanotechnology in woody plant tissue culture, while other regions have made progress. This point alone may not provide sufficient depth or originality for publication. The authors should clarify what specific message or scientific value they intend to convey through this comparison.
-
Reason for the “Nano Gap”: The authors attribute the “nano gap” to a lack of communication and collaboration between nanoscientists and plant biotechnologists. This explanation appears overly simplistic and unconvincing in the current global research environment. A more comprehensive analysis of possible socio-economic, infrastructural, or policy-related factors would strengthen this argument.
-
Awareness in the Balkan Region: The manuscript implies that plant biotechnologists in the Balkan countries may be unaware of developments in nanobiotechnology. This assumption seems questionable and should be supported by evidence or rephrased to avoid generalization.
Author Response
We sincerely thank you for your thoughtful comments, and constructive suggestions. In response, we have carefully revised the manuscript, addressing each of your remarks in detail. All additions are colored in red and blue for your convenience.
Comment 1. Novelty: The authors should clearly articulate the novelty and scientific contribution of this review. As it currently stands, the manuscript primarily summarizes existing research and does not highlight any new perspectives or critical insights.
Response: We thank the reviewer for this helpful comment.
The updated version explicitly identifies the focus on nanotechnology applications in woody-plant in vitro biotechnology across the Balkan region, outlines the comparative and analytical approach, summarizes the main thematic areas discussed, and concludes with a concise perspective on the significance of bridging the “nano gap”.
The abstract has been substantially revised to clarify the objective, scope, and major topics discussed in the review (lines 32 – 35; 44 - 51).
The introduction has been revised. A clear statement is provided in the last paragraph (lines 113 - 122).
A paragraph is revised and added in Section 4 (lines 547 - 551).
The conclusion has been revised. A final paragraph is added to address the above (lines 658 - 662).
Comment 2. Scope and Focus: The review mainly emphasizes that the Balkan region has not yet adopted nanotechnology in woody plant tissue culture, while other regions have made progress. This point alone may not provide sufficient depth or originality for publication. The authors should clarify what specific message or scientific value they intend to convey through this comparison.
Response: We thank the reviewer for this valuable observation. The revised version now clarifies the scientific message and significance of the Balkan–global comparison. Beyond identifying the lack of nanotechnology integration, this paper introduces the “Nano Gap” concept, which links regional scientific readiness, infrastructural capacity, and interdisciplinary connectivity with the effective adoption of nano-enabled in vitro systems. This gives the review broader scientific value: it advances a strategic framework for integration, not merely a regional summary.
All the additions presented in lines 32 – 35; 44 – 51; 113 – 122; 547 – 551; 658 - 662 help to address this concern as well.
Comment 3. Reason for the “Nano Gap”: The authors attribute the “nano gap” to a lack of communication and collaboration between nanoscientists and plant biotechnologists. This explanation appears overly simplistic and unconvincing in the current global research environment. A more comprehensive analysis of possible socio-economic, infrastructural, or policy-related factors would strengthen this argument.
Response: We appreciate this constructive comment. In the revised version, we have expanded the discussion on the causes of the “Nano Gap” to include socio-economic, infrastructural, and policy-related factors, providing a more comprehensive explanation. The “Nano Gap” is therefore reframed not as a purely scientific communication problem, but as a systemic issue shaped by institutional, economic, and policy environments (lines 551 – 559; 563 - 586).
A paragraph is also revised and improved in the conclusion section to address this concern (lines 639 - 645).
Comment 4. Awareness in the Balkan Region: The manuscript implies that plant biotechnologists in the Balkan countries may be unaware of developments in nanobiotechnology. This assumption seems questionable and should be supported by evidence or rephrased to avoid generalization.
Response: We thank the reviewer for this important observation. We agree that implying a lack of awareness among Balkan plant biotechnologists could be misleading and overly generalized. In the revised version, we have rephrased this statement to clarify that the “nano gap” arises not from a lack of awareness or expertise, but from systemic barriers. The updated text emphasizes that Balkan scientists are indeed informed of global developments, but the translation of this knowledge into applied research is constrained by structural and policy-related factors.
To avoid any misunderstanding in this regard, a clarification is added in lines (lines 94 – 100; 560 – 562 in blue color).
Yours sincerely,
Corresponding authors
[On behalf of all authors]

Reviewer 3 Report
Comments and Suggestions for Authors
The manuscript provides a compelling overview of in vitro biotechnology applications for woody plants in the Balkan region. The information presented is relevant and potentially valuable to both academic and industrial sectors, outlining an approach that supports alignment with broader European goals related to sustainable agriculture, biodiversity preservation, and green innovation. Nevertheless, the manuscript requires substantial additions and revisions before it can be considered suitable for publication:
- The abstract should be revised to more accurately reflect the main content of the document. At present, the objective of the review is unclear, as are the key topics discussed. Additionally, a brief concluding statement should be included to summarize the main findings or perspectives.
- In item 2, please add the use of nanoparticles as carriers for genetic material transfer. This is relevant for the development of genetically modified plants with improved traits. For example, gold nanoparticles are commonly used in gene transfer techniques.
- In Figure 2, it would be more appropriate to present a comparison of the status of in vitro plant biotechnology and its nanointegration using quantitative data.
- Key perspectives and challenges for plant nanotechnology should be addressed in the conclusion section: the benefits to plant systems (e.g., propagation of endangered species and improved crop yields), the use of nanomaterials for controlled agrochemical delivery, the mechanisms of nanoparticle uptake and interaction with plant cells, and potential ecological risks, such as environmental accumulation.
Author Response
We sincerely thank you for your thoughtful comments, and constructive suggestions. In response, we have carefully revised the manuscript, addressing each of your remarks in detail. All additions are colored in red and blue for your convenience.
Comment 1: The abstract should be revised to more accurately reflect the main content of the document. At present, the objective of the review is unclear, as are the key topics discussed. Additionally, a brief concluding statement should be included to summarize the main findings or perspectives.
Response: We thank the reviewer for this helpful comment. The abstract has been substantially revised to clarify the objective, scope, and major topics discussed in the review. The updated version explicitly identifies the focus on nanotechnology applications in woody-plant in vitro biotechnology across the Balkan region, outlines the comparative and analytical approach, summarizes the main thematic areas discussed, and concludes with a concise perspective on the significance of bridging the “nano gap.” This revision ensures the abstract accurately represents the structure, content, and conclusions of the manuscript. (lines 32 – 35; 44 – 51 in red color).
Comment 2: In item 2, please add the use of nanoparticles as carriers for genetic material transfer. This is relevant for the development of genetically modified plants with improved traits. For example, gold nanoparticles are commonly used in gene transfer techniques.
Response: We appreciate the reviewer’s suggestion to include a discussion on genetic transformation and nanoparticle-mediated gene delivery. However, the focus of this review is intentionally limited to in vitro micropropagation and preservation systems for woody plants. These two areas represent the core platforms for clonal propagation, germplasm maintenance, and commercial-scale production, which are the most relevant for assessing the current status and practical application of nanotechnology in the Balkans.
Genetic transformation and nanoparticle-mediated gene transfer constitute a distinct research domain with different methodological objectives, regulatory frameworks, and biosafety implications. Therefore, expanding the scope to include transformation studies would dilute the focus of this work. To clarify this distinction, we have added a sentence in the introduction specifying that the review concentrates exclusively on micropropagation and conservation aspects of in vitro biotechnology (lines 106 – 109 in red color). In addition, a brief supporting statement has been included in lines 499–501 (in red color).
Comment 3: In Figure 2, it would be more appropriate to present a comparison of the status of in vitro plant biotechnology and its nanointegration using quantitative data.
Response: We appreciate the reviewer’s suggestion to include quantitative data comparing the status of in vitro plant biotechnology and its nanointegration. However, as noted in the manuscript, there are currently no published studies or datasets documenting nano-assisted in vitro systems for woody plant species in the Balkan region. Consequently, a quantitative analysis of nanointegration is not feasible at this stage. To address this, the capture of Figure 2 has been designed as a conceptual comparative framework that qualitatively illustrates the contrast between the well-established in vitro biotechnology developments and the complete absence of nanotechnology applications in the region (lines 542 – 543 in blue color). Also a sentence is added in the text to clarify the qualitative comparison (lines 534 – 537 in blue color).
We have revised the figure caption and corresponding text in the Results section to clarify this point and prevent potential misinterpretation.
Comment 4: Key perspectives and challenges for plant nanotechnology should be addressed in the conclusion section: the benefits to plant systems (e.g., propagation of endangered species and improved crop yields), the use of nanomaterials for controlled agrochemical delivery, the mechanisms of nanoparticle uptake and interaction with plant cells, and potential ecological risks, such as environmental accumulation.
Response: We thank the reviewer for the valuable suggestion. A paragraph is inserted in the conclusion section (lines 651 – 657 in blue color).
Yours sincerely,
Corresponding authors
[On behalf of all authors]

Reviewer 4 Report
Comments and Suggestions for Authors
Please see the suggestion on the manuscript text

Author Response
We sincerely thank you for your thoughtful comments, and constructive suggestions. In response, we have carefully revised the manuscript, addressing each of your remarks in detail. All additions are colored in red for your convenience.
Comment 1: Applications of nanotechnology in plant tissue culture: Opportunities for the Balkan countries. In this chapter should be included also the applications of nanoparticles in:
- enhancing survival and delaying explant senescence
- in mediation of plant transformation
- in inducing/enhancing somaclonal variation
Response: We appreciate the reviewer’s suggestions and we are explaining as follows:
- We have added as a stand-alone subsection on the use of nanoparticles on Enhancing survival and delaying explant senescence (lines 227–238 in red color). Since the new reference was cited in the added paragraph, we included it in the References section as well (lines 832-833 in red color)
- We appreciate the suggestion to include a discussion on genetic transformation and nanoparticle-mediated gene delivery. However, the focus of this review is intentionally limited to in vitro micropropagation and preservation systems for woody plants. These two areas represent the core platforms for clonal propagation, germplasm maintenance, and commercial-scale production, which are the most relevant for assessing the current status and practical application of nanotechnology in the Balkans. Genetic transformation and nanoparticle-mediated gene transfer constitute a distinct research domain with different methodological objectives, regulatory frameworks, and biosafety implications. Therefore, expanding the scope to include transformation studies would dilute the focus of this work. To clarify this distinction, we have added a sentence in the introduction specifying that the review concentrates exclusively on micropropagation and conservation aspects of in vitro biotechnology (lines 106–109 in red color). In addition, a brief supporting statement has been included in lines 499–501 (in red color).
- We appreciate the reviewer’s suggestion to include the use of nanoparticles inducing or enhancing somaclonal variation. However, since this paper focuses on in vitro micropropagation and germplasm conservation, we did not add this as a stand-alone subsection. Instead, the topic was integrated into subsection 2.2, where we discuss the possible somaclonal variations caused by the use of nanoparticles. In this addition, we highlight that when the goal is clonal propagation, nanoparticles should be applied with caution, whereas in cases where the objective is trait improvement, such induced variation may be beneficial (lines 206–225 in red color). Since the new references were cited in the added text, we included them in the References section as well (lines 822-831 in red color).
Yours sincerely,
Corresponding authors
[On behalf of all authors]

Round 2
Reviewer 3 Report
Comments and Suggestions for Authors
The manuscript “The Balkan Region and the “Nano Gap”: An Underexplored Dimension of In Vitro Biotechnology for Woody Plants” has been improved, and all my questions were taken into account.
I recommend the publication in “Plants”.